# **Exploiting Physics-Based Machine Learning to Quantify Geodynamic Effects – Insights from the Alpine Region**

Denise Degen<sup>1,2</sup>, Ajay Kumar<sup>2,3</sup>, Magdalena Scheck-Wenderoth<sup>4,2</sup>, and Mauro Cacace<sup>2</sup>

**Correspondence:** Denise Degen (denise.degen@gfz.de)

Abstract. Geodynamical processes are important to understand and assess the evolution of the Earth system as well as its natural resources. Given the wide range of characteristic spatial and temporal scales of geodynamic processes, their analysis routinely relies on computer-assisted numerical simulations. To provide reliable predictions such simulations need to consider a wide range of potential input parameters, material properties as they vary in space and time, in order to address associated uncertainties. To obtain any quantifiable measure of these uncertainties is challenging both because of the high computational cost of the forward simulation and because data is typically limited to direct observations at the near surface and for the present-day state. To account for both of these challenges, we present how to construct efficient and reliable surrogate models that are applicable to a wide range of geodynamic problems using a physics-based machine learning method. In this study, we apply our approach to the case study of the Alpine region, as a natural example for a complex geodynamic setting where several subduction slabs as imaged by tomographic methods interact below a heterogeneous lithosphere. We specifically develop surrogates for two sets of observables, topography and surface velocity, to provide models that can be used in probabilistic frameworks to validate the underlying model structure and parametrization. We additionally construct models for the deeper crustal and mantle domains of the model, to improve the system understanding. For this last family of models, we highlight different construction methods to develop models to either allow evaluations in the entirety of the 3D model or only at specific depth intervals.

### 1 Introduction

Present-day manifestations of geodynamical processes such as the formation of mountains and oceans are essential to understand the Earth's system. This system understanding aids, in turn, to better assess the potential and associated risks of exploiting the earth's natural resources. To obtain a quantifiable insight into the factors that influence subsurface processes is a major challenge. The convoluted response of the subsurface processes and the present-day architecture is mainly measured at the surface e.g., by means of geodetic observations, which provide information on near surface conditions over a finite observational time window. In contrast, the underlying processes driving the geodynamic set-up of the Earth system encompass a wider range of spatial and temporal scales. It follows that any study aiming at a systematic understanding of geodynamic processes is

<sup>&</sup>lt;sup>1</sup>Institute of Applied Geosciences, TU Darmstadt, Schnittspahnstraße 9, 64287 Darmstadt, Germany

<sup>&</sup>lt;sup>2</sup>GFZ Helmholtz Centre for Geosciences, Telegrafenberg, 14473 Potsdam, Germany

<sup>&</sup>lt;sup>3</sup>Department of Earth and Climate Science, IISER Pune, India

<sup>&</sup>lt;sup>4</sup>Sediment Basins and Georesources, TU Berlin, Ernst-Reuter-Platz 1, 10587 Berlin, Germany

subjected to several sources of uncertainty. To properly address these uncertainties is a non-trivial task given the diversity of spatial and temporal scales and the tight nonlinearity of the physics at play, resulting in computationally expensive problems to be solved. Therefore, in most studies, the impacts of uncertain parameters, e.g., material properties, and architecture, have been determined either through ("end-member") scenario analysis (Kumar et al., 2022; Lourenço et al., 2020) or uncertainty quantification methods using a low number of forward evaluations (Brisson et al., 2024). Scenario analyses provide a good first-order understanding, but they do not provide any quantifiable uncertainties. To obtain any quantifiable result requires to carry out sensitivity analyses. The caveat here is that numerical investigations of geodynamical processes must tackle problems characterized by a high degree of nonlinearity in the physics and a high dimensional parameter space, which should also include their correlations (Degen et al., 2023; van Zelst et al., 2022). Hence, computationally inexpensive local sensitivity analyses (Saltelli et al., 2019; Wainwright et al., 2014) are usually not applicable. Indeed, not only is the geodynamical model evaluation expensive to forwardly compute, but we in addition need global sensitivity studies (Saltelli et al., 2019; Sobol, 2001; Wainwright et al., 2014) to account for the nonlinearities and correlations. Global sensitivity analyses require numerous model evaluations, typically in the order of 10<sup>5</sup> to 10<sup>6</sup> evaluations. Considering, that a single geodynamical model simulation requires hours to solve, even if assisted by state-of-the-art high-performance compute infrastructures, these analyses thereby become prohibitive (Degen et al., 2023; Sobol, 2001; Wainwright et al., 2014).

To address these computational challenges, we introduce a physics-based machine learning method for the construction of surrogate models. Physics-based machine learning is a rapidly growing field of active research, with physics-informed neural network (PINN; Raissi et al., 2019) as one of its most prominent members. For the purpose of our study, we must emphasize that PINNs restrict the physics through the computation of the loss function, where the physics is considered among other constraints. This entails that PINNs cannot preserve either the full physics at play or the exact input-output relationships (Chuang and Barba, 2022; Raissi et al., 2019). Furthermore, there are reported convergence issues and/or high computational costs for the construction of the surrogate models (Chuang and Barba, 2022, 2023). In this study, we discuss and present an alternative method, the non-intrusive reduced basis (RB) method, a modification of the rigorously proven intrusive reduced basis method (Benner et al., 2015; Hesthaven et al., 2016) to account for nonlinear and hyperbolic partial differential equations (PDE).

The non-intrusive RB method originated from its intrusive version, which is rigorously proven (Benner et al., 2015; Hesthaven et al., 2016). In its original form, the method has been applied to several geothermal and geodynamic case studies (Manassero et al., 2021; Degen et al., 2022; Degen et al., 202

et al., 2016). In its original form, the method has been applied to several geothermal and geodynamic case studies (Manassero et al., 2021; Degen et al., 2021b; Degen and Cacace, 2021; Degen et al., 2022b) including a prior study for the Alpine region focusing on the temperature response (Degen et al., 2021a). However, a disadvantage of the intrusive version, is its inefficiency with respect to applications with a high degree of nonlinearity, as present in this study. To overcome this limitation the non-intrusive version has been introduced by Hesthaven and Ubbiali (2018); Swischuk et al. (2019); Wang et al. (2019) for mathematical studies, demonstrating promising results for nonlinear problems. The method has already been explored for diverse geoscientific applications, such as geothermal and hydrological applications (Degen et al., 2022a, 2023), and for ground motion maps (using a different projection method; Rekoske et al., 2022). Nonetheless, within the field of Geodynamics, the current explorations remain on a benchmark level (Degen et al., 2023). In this paper, we demonstrate the construction of efficient and reliable surrogate models in the context of large-scale nonlinear geodynamical studies and apply our method to the

case study of the Alpine region. To our knowledge, no surrogate models for the velocity and topography exist. We present the construction of the surrogate, associated requirements, and challenges. Note that the case study is chosen to better illustrate the concepts, but the conclusions derived from it are generally applicable.

# 2 Methods

## 2.1 Geodynamic numerical model

All forward numerical simulations presented in this study have been performed with the open source-geodynamic software LaMEM (Kaus et al., 2016). LaMEM is a 3D finite-difference code that uses a staggered grid and a marked-in-cell technique and it is interfaced with the PETSc library (Balay et al., 2017) to solve the resulting nonlinear system.

Regarding the physics of the system, we consider a linear Stokes Flow problem. The governing equations are derived from the conservation of mass and momentum considering a continuum mechanics approximation of the lithosphere and underlying asthenospheric mantle as an incompressible viscous fluid (Einstein summation convention is assumed):

70 
$$\frac{\partial \sigma_{ij}}{\partial x_j} = -\rho g_i$$
, with  $\sigma_{ij} = -P + \tau_{ij}$ , and  $P = \frac{-\sigma_{ij}}{3}$ , (1)

$$\frac{\partial v_i}{\partial x_i} = 0. {2}$$

Here,  $\sigma_{ij}$  denotes the total stress tensor,  $x_j$  the spatial coordinate,  $\rho$  the phase density, g the gravitational acceleration, P the pressure (mean stress),  $\tau_{ij}$  the deviatoric stress tensor, and  $v_i$  the solid velocity. We considered secondary creep as the main deformation mechanism, in which the deviatoric stress is a function of the solid velocity vector via an effective Newtonian viscosity  $\eta$  defined as:

$$\tau_{ij} = 2\eta \dot{\epsilon}_{ij},\tag{3}$$

$$\dot{\epsilon}_{ij} = \frac{1}{2} \left( \frac{\partial v_i}{\partial x_j} + \frac{\partial v_j}{\partial x_i} \right),\tag{4}$$

where  $\dot{\epsilon}_{ij}$  is the deviatoric strain rate tensor. Because of the linearity of the approach no functional behavior of the effective viscosity parameter either in terms of the solid velocity or strain rates are considered.

# 2.2 Non-Intrusive Reduced Basis Method

The non-intrusive RB method is a physics-based machine learning method (Degen et al., 2022a; Hesthaven and Ubbiali, 2018; Swischuk et al., 2019) aiming at the construction of low-dimensional surrogate models that can maintain the input-output relationships. This last feature is what differentiates the non-intrusive RB method from other commonly used physics-based

machine learning methods, such as PINNs (Raissi et al., 2019). Maintaining this relationship is crucial since we want to use the surrogate models as a substitute for the high-dimensional models in the framework of, for instance, a global variance-based sensitivity analysis.

We implement an "offline-online decomposition procedure", where we limit all computationally expensive procedures in the offline stage, which needs to be computed only once. During this stage, the surrogate model is constructed allowing fast evaluations of the surrogate during the online phase. Such an offline-online decomposition is optimal in relation to the requirements of our analysis, which features a many-query and real-time context. Note that many-query refers to an application needing the evaluation of numerous forward simulations, and real-time denotes an application requiring an immediate response.

The surrogate model is a linear combination of weighted basis functions:

100

105

110

95 
$$u_{\underline{red}\,\underline{red}}(x;\mu) = \sum_{n=0}^{N} \alpha_n(\mu) \psi_n(x),$$
 (5)

where  $w_{red}$   $w_{red}$  is the reduced solution, N is the dimension of the reduced model, x is the vector of the spatial coordinates,  $\mu$  the model parameters that are varied (e.g., viscosity and density),  $\alpha$  the reduced coefficients, and  $\psi$  the basis functions. The construction of the surrogate is divided into two stages. First, we compute the basis functions, which capture the general characteristic of the physics, via a proper orthogonal decomposition (POD). During the POD a singular value decomposition is performed. To decide which singular vectors are to be kept, we use the following energy criterion (Guo and Hesthaven, 2019; Swischuk et al., 2019):

$$\frac{\sum_{n=0}^{i} \lambda_n^2}{\sum_{n=0}^{N_n} \lambda_n^2} \le \epsilon. \tag{6}$$

Here,  $\lambda$  is the singular value,  $N_t$  is the total number of singular values, and  $\epsilon$  is the POD threshold. The training data set, containing simulations for different parameter values (e.g., different viscosities and densities), is the input to the POD. Before utilizing the data, they are scaled to ensure a balance of the features, details regarding the scaling are listed in section 4.1. To generate the training data set, we rely on a Latin Hypercube sampling strategy. In this study, it has a size of either 100 or 200 realizations. The weights for the basis functions are determined through machine learning techniques, in our case via a neural network approach. The different parameter values serve as labels, whereas the input data is the matrix product of the training data and the basis functions. Note that an additional validation data set, generated via a random sampling strategy, is also required at this stage. It has a size of 15 to 20 realizations. The different sampling strategies for the training and validation data set are employed to widen the applicability of the surrogate model.

For determining the approximation quality of the surrogate model, we use the mean square error:

$$\frac{1}{N_{\text{data}}} \sum_{i=0}^{N_{\text{data}}-1} (u_{\text{data}} - u_{\text{red}})^2,$$
(7)

where  $N_{\text{data}}$  refers to the size of the training or validation data, and  $u_{\text{data}}$  to the training or validation data realizations.

### 115 3 Case Study: Alpine Region

Quantifying the coupling of the mantle and surface processes is crucial to better understand their roles in landscape evolution and associated hazards. The main challenge to such a quantification by numerical methods stems from the diversity of characteristic time scales at play as well as the variability in the subsurface physical state, e.g., its architecture and physical properties, which translate into high-dimensional and computational costly numerical simulations.

In the Alps, mantle processes in terms of slab dynamics are reported to have occurred in the recent geological past (few Myrs ago; Handy et al., 2021; Fox et al., 2015). In addition, the region also underwent additional surface stressing from the removal of the ice load from the last glacial maximum (~20 Kyrs) to present-day (Mey et al., 2016). Therefore, the region provides us with a natural laboratory to understand the coupling between deep-mantle and surface processes. Recent studies focusing on the Alps and adjacent foreland regions have provided detailed images of both the crustal and mantle lithosphere configuration, allowing to narrow down the source of uncertainties in the architecture of the crust and upper mantle (El-Sharkawy et al., 2020; Fichtner et al., 2013; Kästle et al., 2020; Paffrath and Friederich, 2021; Spada et al., 2013; Spooner et al., 2019; Tesauro et al., 2008; Zhu et al., 2015).

# 3.1 Model setup


In this study, we relied on an available 3D structural model of the crust and the upper mantle together with results from published tomography models to constrain the geometry of the plate in the study area. The model differentiates the main geological
units of interest, that is, from top to bottom: i) sediments, ii) upper-crust, iii) lower-crust, iv) lithospheric mantle, and v) slabs in
the asthenosphere below the lithosphere. The thickness of the crustal layers is taken from a 3D crustal model of Spooner et al.
(2019) complemented by the EuCrust-07 model (Tesauro et al., 2008). The thickness of the lithospheric mantle (i.e., depth of
the thermal Lithosphere-Asthenosphere Boundary; LAB) and slabs in the asthenosphere are taken from Kumar et al. (2022),
which were derived from statistical clustering of available shear-wave tomographic models and their conversion to temperature. In the present study, we limit our choice to the "mean model" of the statistical ensemble, because it has been shown to
be consistent with first-order patterns of observed topography, GNSS-derived vertical velocities, and seismicity in the region
(Kumar et al., 2022).

Reference densities and viscosities in the different layers of the model are chosen according to the representative bulk lithologies and shown in Figure 1. In the crust, they are in agreement with available gravity data, and those for the lithosphere, slabs, and asthenosphere are chosen according to converted temperatures hence consistent with tomography models (Kumar et al., 2022). Effective crustal viscosities are representative of the thermomechanical model of the region (Spooner et al., 2022). In the upper mantle (lithosphere, slabs, and asthenosphere) viscosities are set according to previous studies (Kumar et al., 2022; Mey et al., 2016; Sternai et al., 2019). Density and viscosity ranges are assigned according to the expected variability, arising either from uncertainties or non-uniqueness of these properties. The modeling domain is a 3-D Cartesian box with 96 x 96 x 128 grid points, with a resolution of  $\sim$ 13 km in E-W,  $\sim$ 17 km in N-S, and  $\sim$ 3 km along the depth axis.

**Figure 1.** Illustration of the Alps 3D model and the modeled topography as well as the vertical surface velocities overlain with the measured topography. The parameter ranges for the input densities and viscosites, as well as their reference values, are provided.

The top of the model includes a light and viscous layer of 20 km thickness representing a free-surface through an approach called "sticky-air" (Crameri et al., 2012; Kaus et al., 2010). The models are allowed to dynamically develop topography until a "quasi-isostatic" equilibrium is achieved, and the data is extracted at this time step for the surrogate model construction. Based on the reference model (Table in Fig. 1), we chose this step at 0.27 Myr, at which the topography and velocities evolution stabilizes, and the lithosphere and slabs do not deform significantly.

#### 3.2 Data Sets





To investigate how surrogate models can be efficiently and reliably constructed, we focus on three scenario realizations. In the first scenario, we vary the density values of all layers (exception made for the sticky air) and keep the viscosities fixed at their reference values (density scenario). In the second scenario, we vary the viscosities (except for the sticky air), while setting the densities to their reference values (viscosity scenario). The last scenario is a combination of the previous two, where we vary both viscosities and densities (combined scenario). Ranges of variation for both densities and viscosities are listed in Figure 1. Illustration of the Alps 3D model and the modeled topography as well as the vertical surface velocities overlain with the measured topography. The parameter ranges for the input densities and viscosites, as well as their reference values, are provided.

For all three scenarios, we have to generate a training and a validation data set in order to construct the surrogate model. The training data set for both the density and combined scenario consists of 100 simulations, where the input parameters have been sampled based on a Latin Hyper Cube Sampling (LHS) strategy. We use a quasi-random LHS strategy since it samples more efficiently the parameter space and it avoids clustering that would occur for a purely random approach. Additionally, the LHS technique enables the construction of surrogate models with fewer samples than other strategies, and it does not introduce any source of bias towards specific regions of the parameter space. In an attempt to investigate the sensitivity of the performance as a function of the volume of data, we doubled the dimension of the training data set for the viscosity scenario (200 simulations). To ensure a generic usability of the model, we do not split the data sets generated above into training and validation data. Instead, we generate additional simulations (15 for the density and combined scenario and 20 for the viscosity scenario) to form the validation data sets. Note that these simulations are not generated from the same LHS sampled input parameters but from randomly produced input parameters. The different sampling approaches are chosen to ensure that the constructed surrogate models can be applied to both LHS and non-LHS sampled parameters.

### 4 Results

The goal of this study is to showcase how surrogate models can be constructed in a way to preserve the input-output relationship, hence the physical characteristic of the problem. This is why we focus on the surrogate model construction and discuss how it can be of help for geodynamic applications.

# 4.1 Surrogate Model Construction






In the following, we present the surrogate models for both the topography and vertical surface velocities and their evaluation with emphasis on their capabilities to preserve the physical characteristics of the original problem. In the Discussion section, we elaborate on the advantages and disadvantages of the various surrogate modeling techniques. Note that the hyperparameters for each surrogate model are listed in the tables in Figure 2 and the scripts for the surrogate model construction are provided in Degen et al. (2025).

For an efficient surrogate model construction, we need to pre-process the data. In order to avoid lateral boundary effects in the Alpine region, the simulations have been performed on an extended region Figure 1 following (Kumar et al., 2022). To focus the surrogate only on the Alpine region the extended model volume is removed first. The next steps for the pre-processing are the scaling of the training and validation data sets, as well as the corresponding parameters. For the data sets, we use a min-max scaling, which ensures that the data is between zero and one. This normalization is important because to derive basis function weights using neural networks, we rely on a sigmoid as our activation function, which expects a normalized distribution. For the parameters, we use a z-score scaling, all parameters share a zero mean and a standard deviation of one. Note that for both the min-max scaling and the z-score scaling the minimum, maximum, mean, and variance are calculated for the training data set and parameters, respectively. These values are also used to scale the validation data set and parameters in order to ensure that both data sets are subjected to the same scaling. For the remainder of the paper, we refer to the scaled data sets as the nondimensional data sets, and to the original data sets as the dimensional data sets.

For the vertical surface velocities, one additional pre-processing step is required because the data sets contain entries likely caused by numerical variations and round-off precision errors, that is, entries that do not represent desired physical effects. These entries can vary significantly in their orders of magnitude (e.g.,  $1 \cdot 10^{-12}$  and  $1 \cdot 10^{-16}$ ), affecting optimization via neural network. Hence, we set velocities smaller than  $1 \cdot 10^{-2}$  to zero. While the exact choice of this threshold might be subjective to the modeler and/or motivated by the physical process, this is an important stage to achieve a stable performance during the surrogate model construction.

# 4.1.1 Topography

For the study focusing on the topography (top panels of Figure 2), we define a POD tolerance of  $1 \cdot 10^{-4}$  for the nondimensional data set, that is, we maintain an a global accuracy of 1 m regarding the dimensional data set. This value was chosen because higher accuracies are not possible due to the numerical setup and the potential errors associated with the underlying model assumptions. Note that both the accuracies of the POD and the later error estimators of the neural network (NN) are global error measures, hence errors can be higher locally. This is That is the reason why we should not only consider the global error measure but in addition, investigate the local error distribution for designated realizations of the validation data set. This allows us to ensure that there are no areas of the model that are significantly underrepresented. For a better illustration of these concepts, we have a look at Figure 2. The global errors for the training and validation data set of the model shown in the first panel are in the order of  $10^{-5}$  km<sup>2</sup>, and  $10^{-3}$  km<sup>2</sup>. The mean squared error averages all approximations errors for all locations

in space over the entire set of realizations. Focusing on the deviations between the snapshots and the predicted solution, we obtain maximum error values in the range of 10s of meters (a few percent). Hence, the maximum local variations can exceed the error values indicated by the global metrics.

Figure 2. The top panels show the local error distribution of the topography for the 13th simulation of the validation data set, where the scenario of varying densities is illustrated on the left, the scenario for varying viscosities in the middle, and combined scenario on the right. The middle panels show the same for the local error distributions of the vertical surface velocities. Additionally, in all six panels the global root mean square errors for the dimensional data sets are provided. The lower panels contain the information about the structure of the corresponding surrogate models, including the architecture of the utilized neural networks.

The first observation to make is that both local and global errors are smaller for the viscosity scenario (top right\_middle panel of Figure 2). At first, one might associate this result with the higher amount of training data we chose for this specific viscosity scenario. However, the size of the training data set likely provides only a minor contribution to the observed behavior. Instead, the primary cause of such a behavior stems from the lower complexity of the model, an observation that is supported by a lower number of basis functions required (12 instead of the 20 for the density scenario). A lower complexity is either a consequence of the model setup or the parameters themselves. Regarding the model setup, for the density scenario, we allowed all layers to change individually, keeping only the sticky air layer fixed.

In contrast, for the viscosity scenario we varied again all viscosities (except the one combined several layers in terms of their viscosity parameterization and considered only the viscosity of the lithosphere (including slabs) and asthenosphere as independent parameter. As for the density scenario viscosity of the sticky air layer) but this time only the asthenosphere is changed independently to the other layers is fixed for all realizations. This entails that we have six parameters that are varied for the density scenario, whereas in the viscosity case, we only have two parameters. Another possible reason for the reduction in complexity is that the densities have a higher impact on the present-day topography (via the isostatic principle). This said, by only relying on the number of basis functions required, we cannot make any final concluding statement. In order to confirm or contradict our hypothesis, we would need to carry out a full global sensitivity analysis, which in turn requires, a functioning surrogate model considering both changes for the viscosities and densities. However, as To focus on the conceptual aspects and their implications for geodynamical modeling, we first present the surrogates for the density and viscosity scenarios and we will detail in the surrogate model behavior for the combined scenario in subsection 4.3, this surrogate model is currently not constructable because of challenges related to the data set.





Another observation worth discussing is that the error distribution shows an underlying pattern that does not resemble a random noise distribution. This is caused by the choice of the surrogate model technique. To better illustrate this phenomenonaspect, it is important to note that the non-intrusive RB method can be interpreted as an analog method to a more classical Fourier analysis. In Fourier analysis, the signal is decomposed into different sinusoidal functions, with the first functions cast "low frequency" and the latter "high frequency" information. In our case, we perform a POD to determine the physical characteristics of the problem and disregard those basis functions that do not add a high information content. The top panels show the local error distribution of the topography for the 13th simulation of the validation data set, where the scenario of varying densities is illustrated on the left and the scenario for varying viscosities on the right. The middle panels show the same for the local error distributions of the vertical surface velocities. Additionally, in all four panels the global root mean square errors for the dimensional data sets are provided. The lower panels contain the information about the structure of the corresponding surrogate models, including the architecture of the utilized neural networks.

Back to the analogy with a Fourier analysis, it implies that we do disregard "high frequency" information. Hence, the difference between reduced and full solutions reveals the pattern from the basis functions we disregard.

# 4.1.2 Vertical Surface Velocity






For the vertical surface velocities, we impose a slightly lower POD tolerance of  $1 \cdot 10^{-2}$  for the nondimensional data set, targeting a global accuracy in the order of 0.01 cm/a for the dimensional data set. Consequently, we observe slightly higher errors for the vertical velocities than for the topography. Nonetheless, we obtain trends that are very similar to those discussed for the topography (middle panels of Figure 2). We indeed obtain more accurate surrogates for the viscosity scenario, which, at first guess, we can still associate with a lower complexity. If we take the number of basis functions as a metric informative of the complexity, we should note that for this specific scenario their number increases from the density to the viscosity scenario. This is despite, for the viscosity scenario only two parameters are varied, whereas for the density scenario, six parameters are changed. For the case where all parameters have the same impact, we would expect to have more basis functions for the density case. Given that the opposite occurs, we take this aspect as an indication that the viscosities have a higher influence on the surface velocities than the densities. We can also notice a similar pattern in the local error distribution as for the topography, where the frequency in the error distribution is higher for the viscosity case.

The results discussed so far indicate that we are able to construct a surrogate model that can maintain the physical characteristics of the problem with very few simulations (100 simulations for the density scenario, 200 simulations for the viscosity scenario) when compared to data-driven methods.

# 4.2 Goal-Orientated vs. General-Purpose Surrogate Model

So far we have discussed surrogate models for the topography and the vertical surface velocity as independent properties. However, both properties provide information only limited to the surface conditions. Using and relying on surface properties is essential given that the majority of measurements are gathered there, which we can use to validate the model outcomes. Surface properties are a convolution of the geodynamic makeup of a region and the underlying processes. Indeed, we are often interested in improving our understanding of the underlying driving forces of a system, for which we are required to also consider information from the deeper parts of the model.

In order to construct surrogate models with respect to the deeper domains of interest, it is important to know whether we are interested in the entire 3D volume of the model or only at certain, for instance, depth levels. This is essential since the surrogate models can be tailored to specific "goals", i.e. goal-orientated surrogate model construction. In the following we compare the construction of general-purpose versus goal-orientated surrogate models for the case study of the Alpine region.

Under a general-purpose surrogate model, we aim to understand a model that has no focus on either a specific spatial location or assumes any prior knowledge about the distribution of the input parameters. For our case study, this implies that general-purpose models consider the entire 3D model volume, which differs from the models described above that focus on the surface only. Considering the distribution of the input parameters becomes slightly more complicated since it is not possible to assume any kind of distribution. Ensuring to design a model not only valid for a single distribution (e.g., a normal distribution), we follow the same procedure as described for the surface surrogate models. Hence, we use an LHS method that samples the training realizations from a uniform distribution and a random sampling strategy for the validation set. Note that for the entire

section, we consider the previously described viscosity and density scenarios with the parameter ranges defined in Figure 1. For the goal-orientated models, we construct separate surrogate models for different depth levels (60 km, 100 km, and 200 km). We again consider the same parameter ranges as for the general-purpose models. For each scenario, we develop surrogate models for the velocities in x-, y-, and z-direction, thereby resulting in three general-purpose models for each viscosity and density scenario (in total six general-purpose models). Three times as many goal-orientated models are obtained since we require one model per depth level. In contrast, to the model for the surface velocity, we define a POD tolerance of 1·10<sup>-4</sup> for all models presented in this section. In the following, we exemplarily present the surrogates for the velocity in the x-direction since the surrogates for the velocities in y- and z-direction show a similar behavior in terms of the prediction quality. The associated hyperparameters are listed in Table 1.

First, we compare the approximation quality of the surrogate models at the three different depth levels for the viscosity

**Table 1.** Hyperparameters for the General-Purpose and the various Goal-Orientated surrogate models for the velocity in the x-direction.

|                      | Varying Densities            |                                         |                                          |                                          | Varying Viscosities          |                                         |                                          |                                          |
|----------------------|------------------------------|-----------------------------------------|------------------------------------------|------------------------------------------|------------------------------|-----------------------------------------|------------------------------------------|------------------------------------------|
| Hyperpara-<br>meter  | General-<br>Purpose<br>Model | Goal-<br>Orientated<br>Model (60<br>km) | Goal-<br>Orientated<br>Model<br>(100 km) | Goal-<br>Orientated<br>Model<br>(200 km) | General-<br>Purpose<br>Model | Goal-<br>Orientated<br>Model (60<br>km) | Goal-<br>Orientated<br>Model<br>(100 km) | Goal-<br>Orientated<br>Model<br>(200 km) |
| Hidden layers        | 1                            | 5                                       | 5                                        | 3                                        | 2                            | 3                                       | 2                                        | 2                                        |
| Neurons per<br>layer | 126                          | 137                                     | 144                                      | 159                                      | 204                          | 153                                     | 120                                      | 223                                      |
| Epochs               | 73,269                       | 63,127                                  | 71,869                                   | 69,835                                   | 61,295                       | 52,887                                  | 58,417                                   | 59,723                                   |
| Learning rate        | 1.13·10 <sup>-3</sup>        | 2.83·10 <sup>-4</sup>                   | 4.14.10-4                                | 2.71.10-4                                | 5.89·10 <sup>-4</sup>        | 1.89·10 <sup>-3</sup>                   | 1.07·10 <sup>-3</sup>                    | 3.12.10-4                                |
| Batch size           | 67                           | 83                                      | 64                                       | 72                                       | 89                           | 79                                      | 61                                       | 71                                       |
| Optimizer            | Adam                         |                                         |                                          |                                          | Adam                         |                                         |                                          |                                          |
| Loss function        | Mean Squared Error (MSE)     |                                         |                                          |                                          | Mean Squared Error (MSE)     |                                         |                                          |                                          |
| Activation function  | Sigmoid                      |                                         |                                          |                                          | Sigmoid                      |                                         |                                          |                                          |

scenario, by extracting the corresponding depth slices from the general-purpose model. This is done to allow an evaluation of the approximation quality for the specific depth levels of the general-purpose and goal-orientated surrogate models. Figure 3 shows exemplary results for the velocity in the x-direction. Here, we would like to first draw attention to the number of basis functions. For the general-purpose model, we require eight basis functions to represent the original LaMEM solutions for every parameter combination within given predefined ranges. The number of basis functions does not change for different depth levels since we construct a model for the entire 3D volume. In contrast, the number of basis functions for the goal-orientated surrogate models varies between seven to seventeen, where we find that the number of basis functions increases with depth. This increase in the amount of basis functions is an indication that the complexity increases with depth. It also presents another interesting


aspect since the general-purpose surrogate model has only eight basis functions, whereas the model designed for a depth of 200 km has seventeen basis functions. This might indicate that, potentially, the model for a depth of 200 km can represent the solutions better. The results listed in Figure 3 seem to confirm this idea, since, we obtain lower global errors for the goal-orientated case. For goal-orientated models of the other depth levels, we observe slightly lower global errors in the case of the validation data sets and slightly higher errors for the training data sets. Nevertheless, the differences between the general-purpose and the goal-orientated models are relatively small. Therefore, we can conclude that both models perform equally well in the given case study. Before discussing the results of the density scenario, we should briefly note some drawbacks of global error measures. Throughout the paper, we use the mean squared error to denote the approximation quality. This error measures the average quality at every spatial location for all realizations in the training and validation set, respectively.

**Figure 3.** The first row shows the LaMEM solutions of the viscosity scenario for the velocity in x-direction of the 11th simulation of the validation data set at a depth level of 60 km, 100 km, and 200 km. In the second row, we illustrated the differences between the general-purpose surrogate model and the LaMEM solutions. Analogously, the third row plots the differences between the goal-orientated surrogate models and the LaMEM solutions.

However, in the case that a smaller area is significantly underrepresented with respect to the remaining part of the model, this would not be visible in the global error measure since the former would not significantly contribute to the average values. Therefore, we spatially resolve all results additionally for designated realizations of the validation data set. This allows us to evaluate if certain model areas are significantly worse approximated than the rest, which is not the case for the viscosity scenario.








For the density scenario (Figure 4) the situation changes significantly. We still have a comparable approximation quality between the general-purpose and the goal-orientated surrogate models. However, now the goal-orientated models are slightly worse than the general-purpose models. This comes as a surprise because models designed for the different depth levels should have better global errors than general-purpose models. Furthermore, we observe that despite global errors of the training data set being comparable to the previous scenario, the validation error differs by three orders of magnitude. The question here is: what is the reason for this degraded performance? At first glance, one might associate it with the amount of data. For the previous scenario, we changed two parameters and based the model construction on 200 realizations for the training data set. In this case, we consider the variability of six parameters, while having only 100 realizations for the training data set. As stated previously, this was done on purpose in order to investigate the sensitivity of the surrogate model performance on the amount of data as required for typical geodynamic applications. However, we have discussed in the previous paragraphs how both the viscosity and the density scenario resulted in high-quality surrogate models. Naturally, the data requirement might differ between the surface and the subsurface cases. Indeed we obtain more basis functions for every depth level of the density scenario than for the corresponding cases of the viscosity scenario. Nonetheless, it is unlikely that the data amount is the main and/or sole reason for the degraded performance. If a lack of data is the problem, one would expect that the goalorientated models can partly compensate for this by focusing only on specific model areas instead of the entire model. To further investigate this finding, we lowered the POD tolerance for the goal-orientated model (60 km) to 1·10<sup>-3</sup>. This reduced the amount of basis functions from 29 to 9 and consequently improved the ratio of available data to trainable weights. The obtained approximation errors for the validation data set are comparable though slightly worse, and we still remain with a discrepancy between the training and validation data set. The first row shows the LaMEM solutions of the viscosity scenario for the velocity in x-direction of the 11th simulation of the validation data set at a depth level of 60 km, 100 km, and 200 km. In the second row, we illustrated the differences between the general-purpose surrogate model and the LaMEM solutions. Analogously, the third row plots the differences between the goal-orientated surrogate models and the LaMEM solutions.

Focusing on the first row of Figure 4, we observe an increase in velocity with depth more pronounced than for the viscosity scenario. Furthermore, by investigating the approximation quality of all realizations of the validation data set, we observe that some realizations are extremely well-fitted, while others show significant deviations. At the same time, changes in velocities as well as their maximum magnitudes are significantly higher than for the viscosity scenario. For instance, the maximum velocities in the x-direction are around 1 cm/a for the viscosity case and over 5 cm/a for the density case. This is partly caused by the circumstance that the driving forces are constant in the viscosity scenario because of the constant densities. However, the difference in the obtained velocity patterns and magnitudes cannot might not solely be related to the governing equations and might be caused by the chosen numerical implementation (we discuss this aspect in detail in the Discussion section). This

is most pronounced for the surrogate models representing the deeper model parts but also occurs for the surface models, we jointly discuss potential causes in the next sections. Regardless of the reasons for these differences, any change that cannot be related to the physical characteristics has a high chance to degrade the surrogate model quality.

# 345 4.3 Combined Scenario of Changing Densities and Viscosities







So far we have been discussing the surrogate model construction for either the density or the viscosity scenario. Naturally, it would be is interesting to enable the possibility of changing both the densities and viscosities simultaneously to allow for investigations of parameter correlations. Consequently, we construct surrogate models for the topography (top right panel Figure 2) and vertical surface velocity (middle right panel Figure 2), following the same procedure as detailed in Section 4.1.1 and 4.1.2.

The surrogate model for the topography exhibits a similar global approximation quality than the surrogate model for the density case. With 24 basis functions it is also dimension-wise closer to the density than the viscosity scenario. The observations for the surrogate model of the surface velocity are similar. The number of basis function of the density and combined scenario are identical and the global errors for the validation data set are in between the density and viscosity scenarios. Worth to note is that scenarios involving changes of the densities have overall a degraded approximation quality.

To better understand the cause of this behavior, we have a closer look at the training data sets. In Figure 5 and Figure 6, we present three realizations, as well as the computed entropy for the three scenarios of the topography and vertical surface velocity, respectively. For the combined scenario The entropy is of interest since it illustrates the variability within the data set, which is the key metric for the surrogate model complexity and quality.

Focusing first on the topography (Figure 5), we obtain unrealistically low heights. The computed entropy reveals that one of the most affected model areas is the Alps domain. This is critical since this is our primary area observe similarities for the scenario of varying densities (first row of Figure 5) and the combined scenario (third row of Figure 5) both regarding the pattern and the values of the information entropy. For the varying viscosity case the minimum entropy shows a localized curvilinear pattern matching the geological features of interest in terms of modeled topographic elevations. For the vertical the study area, e.g., transition from the Alpine mountains to their foreland. In addition, the information entropy values are lower illustrating less variability in the data set for the varying viscosities scenario. This is also observable for the surface velocities (second row of Figure 6), we obtain extremely high uplift rates, which are in clear contradiction to the low values obtained. We do observe a localized low entropy pattern, similar to the one discussed for the topography for the associated realization. Again, the entropy illustrates how our area of main interest is subjected to pronounced variability in the velocities, which exceeds our expectations. This abnormal behavior is found for in the varying viscosity scenario (Figure 5) for both the topography and vertical surface velocity only associated with the combined scenario. The reasons behind this behavior will be discussed further in the next section. However, because of the inconsistency in varying viscosities and combined scenarios (second and third rows of Figure 6). An important observation is that the entropy values are lowest in the viscosity scenario for both the topography and velocities, while all scenarios involving density changes tend to produce higher entropy. In addition, the spatial distribution does also differ, being diffuse for cases where we vary the density compared to the cases where viscosity is varied, which tends to be more localized. The difference in the pattern stems from the driving force at play, which is constant for the data, we decided not to construct surrogate models for the combined scenario since a retrieval of a plausible model based on the current data sets is not possible. viscosity scenario (density does not change) and varies for the density scenario. This also applies while discussing lateral variations in the responses, which are also caused by lateral variations in the layer thicknesses. In other words, while varying only the viscosities implies to have models that differ only in terms of the system response to the same set of driving forces, varying also the density add an additional degree of freedom to the problem where not only the system response varies but also the driving force. This feature can explain the differences observed in the entropy patterns and magnitudes observed for the different cases. Generally, we can consider a lower variability as an indication of a lower complexity of the data thereby yielding better approximation qualities, as we observe in the present study. Nevertheless, it is important to discuss (see Section section 5) what are potential causes for the encountered data spread and if they solely reflect desired effects.

**Figure 4.** The first row shows the LaMEM solutions of the density scenario for the velocity in x-direction of the 11th simulation of the validation data set at a depth level of 60 km, 100 km, and 200 km. In the second row, we illustrated the differences between the general-purpose surrogate model and the LaMEM solutions. Analogously, the third row plots the differences between the goal-orientated surrogate models and the LaMEM solutions.

### 5 Discussion

We have obtained promising results for the surrogate models of both the topography and vertical surface velocity for the viscosity and density all scenarios. For the topography, we obtain errors in the order of 10s of meters (density and combined case) or in the order of meters (viscosity case), and for the surface velocity, the errors are below 1 mm per year. In both cases all three cases, the measurement and conceptual errors are much higher, that is, the surrogate models are equally representative than as the full-order models. At the same time, we obtain a major computational gain. The full-order solution for a model takes around 8 hours on the high-performance compute cluster JUWELS using 16 cores. In contrast, the evaluation of the surrogate model requires only 2 to 3 ms (on a Dell Precision 7680, Intel Core i7 Processor, 32 GB RAM), yielding a speed-up of  $1 \cdot 10^7$ .






390

For the sake of clarity, we should note that the gained speed-up does not include the computational cost of the offline stage. We, therefore, open a discussion in the following paragraph where we elaborate on types of applications for which the development of surrogate models is likely to be meaningful. Surrogate models are most beneficial in the for so-called many-query context or for real-time applications. Many query applications, that is, study cases that require to perform case studies that require performing numerous model evaluations, are of specific interest for the presented case studies. One of these applications is a global sensitivity study, which requires many model evaluations (Sobol, 2001), reaching for basin-scale applications hundred thousands of evaluations or more (Degen and Cacace, 2021; Degen et al., 2021b, a). The main factor contributing to the computational cost for the construction of the surrogate models is the generation of the data sets. For the density and combined case, we computed in total 115 simulations (and 220 for the viscosity case). A secondary high cost is. High cost is also associated with the hyperparameter tuning. This costhowever, however, can be significantly reduced by employing a Bayesian optimization with hyperband algorithm (Falkner et al., 2018). Furthermore, we only train for the weights of the basis functions and, not for the entire solution, meaning that typically simple architectures of neural networks are sufficient. Computational times for hyperparameter tuning can differ depending on the tested hyperparameter ranges, but typically do not exceed a couple of hours, a computational time similar to a single simulation. All other costs are orders of magnitude smaller. Hence, the total cost can be well approximated with by the cost of the data set generation. It follows that the method pays off already for a couple of hundred solves, which is orders of magnitude smaller than the amount of evaluations associated with global sensitivity analysis or uncertainty quantification (Degen et al., 2021a, 2022a, 2023).

In this study, we did not perform tests to construct the surrogate for the viscosity case with less training data. Based on the overall performance of the model and the higher amount of model parameters that were varied in the case of the density scenario, it is likely that already slightly more than a hundred simulations would have been sufficient for the viscosity scenario. The approach presented in this study requires orders of magnitude less runs than alternative data-driven machine learning models (Degen et al., 2023; Santoso et al., 2022). Because of this relatively low amount of required simulations, the non-intrusive RB method makes computational computationally expensive analyses feasible for geodynamic applications, enabling a quantification of the uncertainties and correlations.

420 Another advantage of the non-intrusive reduced basis method is that it constructs interpretable surrogate models. This feature

**Figure 5.** Overview of the different distributions of the topography for the models with varying densities, varying viscosities, and varying densities and viscosities. In addition to three realizations per model, we plot the entropy (which captures the average uncertainty/variability of a given data set) to illustrate the variations for all realizations of the individual models. Note that the magnitudes of color scale are different for the three scenarios to better illustrate the pattern of the distributions.

sets apart the method this method apart from other physics-based machine learning methods such as PINNs (Raissi et al., 2019). We should also add that we rely on so-called "black-box methods" only for the projection step. This entails that in our case, non-rigorously provable errors are at most a linear combination of scalar factors affecting only the weighting of the basis

**Figure 6.** Overview of the different distributions of the vertical surface velocities for the models with varying densities, varying viscosities, and varying densities and viscosities. In addition to three realizations per model, we plot the entropy (which captures the average uncertainty/variability of a given data set) to illustrate the variations for all realizations of the individual models. Note that the magnitudes of color scale are different for the three scenarios to better illustrate the pattern of the distributions.

functions but not the entire solution.

In this study, we also presented two ways of constructing surrogate models, whether general-purpose or goal-orientated. For the application presented in this study, we found that the general-purpose model already represented all three depth levels of

60 km, 100 km, and 200 km, so there is no need for a goal-orientated design. Despite our specific results, we briefly discuss the general advantages and disadvantages of this approach. A general-purpose model (as presented in this manuscript) has the advantage of being applicable to the entire model domain. This is not possible for goal-orientated models since they are trained with only the data of the respective area of interest (depth level specified depth levels in our study). However, if the behavior of one goal-orientated surrogate model exceeds the behavior of the general-purpose model it is desirable to have a goal-orientated approach to ensure that the local characteristics are maintained.








This said, one should be also also be aware of the potential increase in computational cost if relying on a goal-orientated approach. As an example, in the application presented in our study, we are interested in three different depth levels yielding three times as many surrogate models as for the general-purpose setting. Since the same data sets can be used for all models, just extracting different information each time, the main costs for both settings are comparable. Also, the time required for the hyperparameter tuning is similar for one general-purpose and one goal-orientated model. This is a completely different situation than for data-driven approaches. Focusing on a single depth level means considering fewer spatial nodes, which would, for a data-based approach, result in a reduced training time. For the non-intrusive reduced basis method, we first perform the POD and then determine the weights of the obtained basis functions through an-NN. This means that our "labels" are still viscosities and densities, but the training data is the product of the snapshots and the basis functions. Consequently, the training data for the NN has the dimension of the number of snapshots times the number of basis functions. Hence, the amount of considered nodes does not impact the computational cost of the NN. This entails that we spent three times as much resources for the hyperparameter tuning of the goal-orientated models than for the general-purpose models. Nevertheless, this cost increase is in the order of a couple of hours and can be done for several models in parallel.

It is also viable to combine both settings by generating a custom loss function for the general-purpose model. Through this custom loss function, it is possible to assign higher weights to specific features (our tested depth levels). This entails that the resulting model can perform better than a pure general-purpose model at these specific depth levels, while still being valid for other parts of the model.

inconsistencies were also observed for the density scenario focusing on the subsurface. Considering the randomness of the observed numerical instabilities in terms of the parameter space, the problems are likely associated with internal numerical aspects, including, for example, the type of advection scheme used, the stabilization of the free-surface (sticky-air), or a combination of both. To test that the observed instabilities are associated to the specific numerical settings of the forward simulation. The density and combined scenarios yield a degraded surrogate model approximation quality, which is especially pronounced for the subsurface models. This is somehow counterintuitive, since the governing equations indicated a low-dimensional parameter space and behavior, conditions that are favorable for the proposed technique. We must note that all forward geodynamic models were allowed to run for 0.27 Myr to achieve a quasi-isostatic balance, hence they represent a snapshot of the transient evolution of the system. For this representative time step, both the topography and surface velocities are stable, but the deeper

A last note on the observed instabilities (Section 4.3). These were especially evident for the combined scenario, though

dynamics (e.g., slabs) might not represent a stabilized state (i.e., equilibrium). This could explain the relatively degraded

Figure 7. Example of the velocities in the entire model domain computed for two of the models from combined scenario data. Panels (left and middleright) shows the distribution of vertical velocities (light-blue) computed using ASPECT employing single advection and stokes (SA-SS) advection scheme. Overlay plotted (orange) are the corresponding vertical velocities from LaMEM. Note the magnitude of the computed velocities. Right panel shows the comparison of vertical velocities for model 82 where single advection and single stokes schemes (same as in middle panel) except that for the orange case we add the sticky air as a physical layer () in the model with a free-slip boundary condition at top.

subsurface surrogate models for the velocity. A potential solution would have been to let the system running over a longer time. However, to run the model longer in time would have triggered additional deformation in the deeper components (e.g., slabs), which would have hindered to approach a state of isostatic equilibrium under the present-day state configuration. The initial phase of the model evolution (i.e., isostatic balance) is primarily controlled by resolved density contrasts, which in our case lies mainly across the boundary between the crust and mantle, closer to the surface. By varying the density configuration, we do modify not only the system response, but also the driving force, while the viscosity only affects the system response (in time) to the same force set-up. This could explain the well-behaved surrogate models of the viscosity-only scenario where densities (i.e., driving forces) are not varied. This aspect could also explain the well-behaved surrogate models of the topography rather than the velocity, which could be more sensitive to the deeper mantle dynamics. As discussed above, we could have allowed the models to run for longer time e.g., 1Myr in order to alleviate this behavior. However, that would have led to significant deformation of the mantle structure and slabs' geometry, and would not represent the present-day state. While in the present study we fixed the geometry of the models future studies will focus on surrogate models that also consider the time evolution of the geodynamic processes.




We would like to conclude with a short note on the flexibility of the surrogate model construction. The proposed methodology is not restricted to a specific forward solver or a specific numerical setup. To illustrate this, we conducted a secondary investigation relying on a different geodynamic software, ASPECT version 2.5 (Heister et al., 2017; Kronbichler et al., 2012)

ASPECT stems from the fact that its core kernels are different from those implemented in LaMEM (finite elements instead of staggered finite differences) and that ASPECT provides with additional flexibility while dealing with particle advection as well as the free surface. We note that, for the combined scenario, the vertical velocities for the two sample models using ASPECT are orders of magnitude lower compared comparable to those obtained in LaMEM (Figure 7). We find additionally that adding a low viscosity and lighter sticky air in the ASPECT models significantly increases these instabilities in velocities (). By testing different advection schemes available in ASPECT, we found that they all give very similar solutions. Thus overall, the results obtained with ASPECT were more stable and closer to the expected solutions than those obtained with LaMEM. That being said, our conclusions should not be perceived as a judgment about which software performs best. We appreciate that both discussed softwares LaMEM and ASPECT are high-level tools and extremely useful to the community. Our aim is to provide the base to objectively discuss our findings in an open manner with the interested scientific community, also The similar velocities obtained by the different forward solvers would entail similar data sets, illustrating the flexibility of the approach with respect to any suggestions. The instabilities are a consequence of the relatively complex geometry and model set-up, showcasing the importance of the development of more realistic benchmarks to further test current software implementations the forward solver used.

### 6 Conclusions





To conclude, we are able to present a method to construct reliable and physically consistent surrogate models. These models are generated with data sets constructed out of 115 simulations for the density scenario (and the scenario for considering the variability of both the densities and viscosities), and 220 simulations for the viscosity scenario. This means we obtain high-quality surrogate models at a cost that is affordable even for computationally demanding applications such as those often encountered in geodynamic applications. Through these surrogate models, we reduce the computational cost from hours on an high-performance compute cluster to milliseconds on a local machine, while maintaining a similar accuracy than the LaMEM full-order forward simulations. Consequently, we are able to capture the present day quasi steady-state for any given density and viscosity values within the pre-defined training ranges.

The presented surrogate models are constructed based on the provided simulations. This has the advantage of not requiring any measurement data for the construction phase. This is crucial since in the given application data could be sparse e.g., mostly available at the surface in case of geodynamic application, and is required for validation purposes of the model. However, this also means that any challenges associated with the forward simulations, such as the presented advection scheme, affect the ability to construct meaningful surrogate models affecting the follow-up quantitative analysesFurthermore, because of the non-intrusive nature of the approach, it is independent of the forward solver. This makes it attractive since it can be interfaced to all existing code packages developed within the computational geodynamic community.

Because of the challenges encountered with the numerical setup, we We focused the paper entirely on the surrogate model

construction for both illustrating the great in order to illustrate the potential this method offers and highlighting to highlight and discuss current challenges and pitfalls. Future studies will investigate the dominant lithological layers in terms of their densities and viscosites viscosities through global sensitivity analyses using the presented surrogate models. Additional, it would be interesting to extend the current approach to capture the dynamical evolution of the system. Transient systems have been considered before both from the framework development (Wang et al., 2019), and for geoscientific applications (Degen and Cacace, 2021).

Code and data availability. The training and validation datasets, their associated model parameters, and the non-intrusive RB code for the construction of all surrogate models are published in the following Zenodo repository (https://doi.org/10.5281/zenodo.17034124, Degen et al., 2025). The input files and geometries for LaMEM and ASPECT are available through the following Zenodo repository (10.5281/zenodo.17051562, Kumar, 2025a). Scripts to generate input files and to extract data for the surrogate models, and LaMEM input files for all the scenarios are available through Zenodo repository (https://doi.org/10.5281/zenodo.16640814, Kumar, 2025b). LaMEM (Kaus et al., 2016) is an open-source code and the used version can be downloaded from https://doi.org/10.5281/zenodo.7071571 or https://bitbucket.org/bkaus/lamem/src/master/. ASPECT version 2.5.0, (Heister et al., 2017; Kronbichler et al., 2012) used in these computations is freely available under the GPL v2.0 or later license through its software landing page https://geodynamics.org/resources/aspect or https://aspect.geodynamics.org and is being actively developed on GitHub and can be accessed via https://github.com/geodynamics/aspect or Zenodo (https://doi.org/10.5281/zenodo.8200213).




Author contributions. All authors discussed and interpreted the presented work. A.K. carried out the forward simulations in LaMEM that served as the training data for the surrogate, as well as the additional ASPECT simulations. D.D. constructed the surrogate model, the analyses of these models and drafted the paper.

530 *Competing interests.* Mauro Cacace is a member of the editorial board of the journal Geoscientific Model Development. The authors declare that they have no conflict of interest.

Acknowledgements. This work is funded by the Deutsche Forschungsgemeinschaft (DFG) through DEFORM project within "Mountain Building Processes in Four Dimensions (4DMB)" SPP (SCHE 674/8-1). The authors acknowledge the Earth System Modelling Project (ESM) by providing computing time on JUWELS cluster, Jülich Supercomputing Centre (JSC) under the application 24312 titled "Unconventional exploration methods for complex tectonic systems-linking geology and geodynamics via numerical modelling". Furthermore, D.D. was supported through funding provided by the Bundesministerium für Bildung und Forschung through the project PBML-HM (project number: 01lS24062).

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
