# Peer review of "Exploiting Physics-Based Machine Learning to Quantify Geodynamic Effects – Insights from the Alpine Region"

_EGUsphere, 2025_

## Community Comment (CC1)

**Performing forward simulations with a free surface using ASPECT & LaMEM: comments on Degen et al. (2025)**

*Boris Kaus, Johannes Gutenberg University Mainz - July 8, 2025*

Degen et al. (2025) submitted a manuscript to GMD in which they use a setup based on data of the Alps to simulate the surface velocity and topography that is expected to develop from a layered lithosphere (Degen et al., 2025). The setup involves several layers with linear viscosity and density below a free surface. The results of forward simulations are used to train reduced basis method surrogate models, and a series of simulations are performed in which 1) the density of the layers was varied, and viscosity of the model was kept constant, 2) the viscosity of the layers was varied while the density was kept constant and 3) both viscosity and density are varied.

Whereas model series 1) and 2) gave forward simulations that appear reasonable to Degen et al. (2025), they report large surface velocities in some simulations of series 3). Without having an exact, or independent solution, Degen et al. (2025) conclude that the large magnitude of velocities is likely caused by "numerical instabilities" of the employed software, LaMEM (Kaus et al., 2016). To further elaborate this, Degen et al. (2025) performed simulations using the software ASPECT (Heister et al., 2017) and report very much smaller vertical velocities from LaMEM, which they interpret to be further evidence that there are instabilities in LaMEM.

The difference of the results from ASPECT and LaMEM could be related to the fact that ASPECT uses a free surface condition that is natural to finite element formulations. In contrast, LaMEM uses a "sticky-air" approach with internal surface to approximate the free surface, as that is more advantageous for structured finite difference methods. Therefore, to quantify the discrepancies of the two methods and evaluate the potential magnitudes of "numerical instabilities" (if any), I reproduce some of the forward modelling results of Degen et al. (2025) and provide more details below. The results clearly show that the discrepancies between ASPECT and LaMEM were caused by the incorrect configuration of the numerical ASPECT models in Degen et al. (2025). Perhaps the most important reason for the discrepancies is that Degen et al. (2025) seem to have placed all the materials upside down in their ASPECT simulations. A second issue is that surface velocity fields extracted from some of the simulations were taken at a stage that the models were not yet in isostatic equilibrium. Since these results were used to train a reduced basis method, this raises significant doubts about the validity of the results of Degen et al. (2025). The inappropriateness of the initial configurations used by Degen et al. (2025) could have been spotted upon inspection and plotting of their results.

Below I proceed with a detailed examination of the various models and with comparing both ASPECT and LaMEM results. The discrepancies are also discussed. Finally, this study shows the importance of detailed code documentation, together with uploads of input files, that allows for reproducibility of scientific results.

**1.  Reproducing the ASPECT & LaMEM results of Degen et al. (2025)**

Since both ASPECT and LaMEM pass the same free surface benchmarks (Crameri et al., 2012; Kaus et al., 2010; Rose et al., 2017) as part of their testing suites, the reported significant differences in results for a simple setup with linear viscosities is rather surprising. It is therefore interesting to reproduce their results using the ASPECT and LaMEM input files provided in (Kumar, 2025). The accompanying README file indicates that ASPECT simulations were performed using ASPECT version 2.5.0. Yet, the provided input file did not work with ASPECT v.2.5.0 since the output parameter "surface elevation" was only introduced in version 3.0.0 (see https://github.com/geodynamics/aspect/pull/5563 ). We have therefore commented this out, after which the free surface ASPECT simulation setup worked with v2.5.0 as provided through a Docker image (see https://aspect-documentation.readthedocs.io/en/latest/user/install/docker-container/installing-docker.html ). In the sticky air layer setup, we additionally had to comment out the line "set Mesh deformation boundary indicators = top: free surface" since a free slip upper boundary was used.

An additional issue with the manuscript and data files is that it is unclear how the model parameters of the input script "alps_HR_84.prm" are linked to the density /viscosity values listed in the file "TrainingParameterAlps_LHS_100_vary_viscosity_density.txt". The parameters used in "alps_HR_84.prm" correspond to those listed on line 84 of "TrainingParameterAlps_LHS_100_vary_viscosity_density.txt", but since the first line is a comment, it appears to actually be model 83 of that series. As will be discussed below, the same confusion remains for other simulations.  We can thus not state with certainty which parameters were employed in the various figures shown in the manuscript (a clear numbering in the data tables would have been very helpful).

To start, we ran the ASPECT setups as provided. To compare this with LaMEM, we employed the same density and viscosity values for the LaMEM setup. Since only the marker input files were provided for the LaMEM setup (and not the MATLAB or Julia input file to generate them), we can only run this setup on 16 cores with a fixed resolution of 96x96x128 cells.  The ASPECT simulation is performed at a much lower resolution of $16^3$ elements (4096 elements in total). Those are, however, quadratic elements and in the VTK/Paraview visualization, $32^3$ elements are shown. To run the LaMEM simulations, we employed precompiled binaries of LaMEM and PETSc as provided through the LaMEM.jl package (version 0.4.7). We slightly

modified the options to switch off nonlinear iterations which are not required in this case as the setup consist of linear viscous materials only.

Results are shown in figure 1. Clearly, there is a significant difference between the initial model configurations. Whereas the LaMEM setup has a layered lithosphere directly underneath the free surface, the ASPECT simulations have the initial model setup flipped upside down! The lithosphere is thus lying upside down at 400 km depth, directly above the no-slip upper boundary. This still results in motion of the free surface, but the magnitude is much smaller given the distance of the density anomalies from the free surface and the viscosity of the lithosphere. The fact that the initial setup is upside down is even clearer if we run the simulations forward in time which results in blobs of sticky air rising through the mantle in the ASPECT setup with sticky air (Fig. 1).

To test whether this was simply a mistake with uploading the input setup to Zenodo, or an actual mistake in the manuscript, we made an attempt to reproduce the histogram of Fig. 7 of the manuscript, which shows the vertical velocity in the full model domain for different ASPECT and LaMEM simulation cases. Since the manuscript does not explain *for which timestep* the vertical velocities were extracted, we performed a number of simulations in which we used the input parameters of lines 82,83,84 and 89,90 of the file.

"TrainingParameterAlps_LHS_100_vary_viscosity_density.txt" as input parameters. The best match with the histogram on fig. 7 was obtained if we use a very early timestep of the simulation along with the parameters provided in lines 90 and 83 of the input file (which would in that case correspond to realizations 82 and 89 in the manuscript). Timestep 0 of LaMEM simulations is typically the initial guess which employs a constant viscosity throughout the whole model domain and should thus not be used, which is why we show timestep 1.

If we use the results very early on in the simulations (timestep 0 for ASPECT and timestep 1 for LaMEM), the resulting velocities and velocity distributions are similar in magnitude as the ones shown on Fig. 7 of Degen et al. (2025) even though the absolute values of the y-axis of the histogram are different (likely because a different normalization was used). Cross-sections of vertical velocities through the model domain of the different setups confirm that this is consistent with an upside-down model setup (see Figure 2). Velocities at the free surface are much smaller in the ASPECT models because the major density anomalies are located far away from the free surface (and in the vicinity of the no slip lower boundary).

Next, we corrected the input model setup of the ASPECT models and repeated the simulations. In this case, there is a quite good agreement between the ASPECT models with a free surface and the LaMEM models, particularly given the large difference in numerical resolution employed (Fig. 3).

[Figure]

**Figure 1.** Temporal evolution of 3D models of the Alps performed with both ASPECT and LaMEM, as used in the work of Degen et al. (2025). An initial flat topography is employed, with either a true free surface for ASPECT (top row) or with a sticky air layer for ASPECT (middle row) and LaMEM (bottom row). The LaMEM setup shows an initial isostatic balancing stage during which topography builds up, followed by a stage that is dominated by the sinking of the denser mantle lithosphere in the asthenosphere. Clearly, the initial ASPECT model setup is flipped upside down, which results in collapsing of the mantle lithosphere and by the development of plumes (or Rayleigh-Taylor instabilities) of sticky air that rise upwards through the mantle. Note that the ASPECT simulations employ a resolution of $32^3$ (quadratic) elements whereas the LaMEM simulations employ a resolution of 96x96x128 staggered FD cells. See text for further details. The simulations were performed by using the files "alps_HR_84.prm" and "alps_HR_84_air.prm" in (Kaus, 2025), taken from (Kumar, 2025) but with correcting typos such that the files are compatible with ASPECT v2.5.0. The LaMEM simulation employs the setup "alps_lith_slabs_400km_HR_reg_HR_84_air.dat" provided with the same density and viscosity values as in the ASPECT setup.

[Figure]

**Figure 2.** Histograms of vertical velocity throughout the model domain for ASPECT and LaMEM simulations for model parameters provided on line 90/83 of the parameter file. Absolute velocities of the LaMEM and ASPECT are comparable to those shown on Fig. 7 of Degen et al. (2025), confirming that an upside-down setup was used to generate the ASPECT models. Also shown are vertical cross-sections through the center of the domain with vertical velocity, which shows major differences in velocity pattern and magnitude.

[Figure]

**Figure 3.** As figure 2, but for cases in which we flipped the initial model setup of the ASPECT models in the vertical direction to make it consistent with the LaMEM (correct) initial model setup; the ASPECT files "alps_HR_83_correct.prm" and " alps_HR_90_correct.prm" in (Kaus, 2025) are used for this. Now there is a reasonable agreement between ASPECT models with a free surface and LaMEM simulations, even though the ASPECT models are performed at a considerably lower resolution. The largest differences occur for the ASPECT sticky air models (lower row) vs. the ASPECT free surface models (middle row), which might be because the sticky air layer is only resolved in the vertical direction by a single (quadratic) element.

The conclusion of this comparison is that the Degen et al. (2025) indeed used an upside-down initial model setup for the ASPECT simulations to compare it with a correctly oriented LaMEM setup. It is of course not a big surprise that significant differences occur, which disappear if both setups are correctly oriented.

What was made clear in the present analysis is that both ASPECT and LaMEM provide similar results for the same initial configurations. In the results observed by Degen et al. (2025), without visualizing them, they stated that:

*"Considering the randomness of the observed numerical instabilities in terms of the parameter space, the problems are likely associated with internal numerical aspects, including, for example, the type of advection scheme used, the stabilization of the free-surface (sticky-air), or a combination of both." (line 385)*

*"Thus overall, the results obtained with ASPECT were more stable and closer to the expected solutions than those obtained with LaMEM." (line 396)*

*"The instabilities are a consequence of the relatively complex geometry and model set-up, showcasing the importance of the development of more realistic benchmarks to further test current software implementations" (line 402)*

Clearly, the discrepancy in the results of Degen et al. (2025) were not the result of lack of realistic benchmarks, but they rather reflect a lack of scientific rigor and self-critical inspection of their own results.

**Geodynamic simulations involving a free surface**

Degen et al. (2025) report "numerical instabilities" with LaMEM in certain model cases, as discussed in section 4.3, which occurs in cases with variable density and viscosity and not in cases with either variable density or variable viscosity. In those cases, they obtain "unrealistically low heights" and "extreme uplift rates" which are "... in clear contradiction to the low values obtained for the topography for the associated realization...." (line 316-317).

Another important issue at hand is that, based on the reasoning of Degen et al. (2025), the LaMEM simulations are considered correct in some cases but must be wrong in other cases based on some perception on what velocities "should be". In the absence of a true solution (e.g. analytical solution), one cannot trust the result in only half of the cases and ignore the other ones. If the study of Degen et al. (2025) suggest that the code is wrong, then its use on training the reduced order models is not justified and all the training set needs to be re-examined.

Velocities in geodynamic models can certainly be very large if one uses small values of viscosities or large density differences. A simple scaling analysis states that the relevant vertical velocity, $v_z$, of a perturbation with amplitude $A$ and density difference $\Delta\rho$ scales as $v_z \sim \Delta\rho g A/\eta$ where $\eta$ is the viscosity of the material.

[Figure]

**Figure 4.** Illustration of a thought experiment in which an ice-cube is pushed down to the water level (A). If the finger (or load) is removed, it will rapidly rise to reestablish isostasy (B). Melting of the ice will cause its elevation to go down with time which happens on a much longer timescale (C). The initial geodynamic model setup used in Degen at al. (2025), starts with a low-density lithosphere that has an initially zero topography. Likewise, the simulations first have to re-establish isostatic equilibriums before the geodynamic effects become visible.

What seems to be the issue here is another factor, which is that the initial setup has lithospheric density anomalies but an initial zero topography. This model is very far from isostatic equilibrium. To illustrate the importance of such a setup, we will consider a very intuitive analogue (Fig. 4). This analogue setup is the case of loading an ice cube just underneath the surface in a glass of water (see figure above, stage A). If the load is removed, the ice cube will rise rapidly, until it reaches the well-known isostatic formula for ice in water (B). After this initial stage, the ice cube will start melting which will slowly lower the elevation of the top of the ice-cube with time (C). In this case, melting the ice cube is used to emphasize that density redistribution will lead to the relaxation of the topography. The first stage (from A to B) is an artefact of the initial model configuration that does not guarantee isostatic equilibrium at time zero ("pushing the ice-cube down"). The second stage (from B to C) is the one we are interested in, if we wish to understand how dynamic processes affect the surface topography (of either the ice cube or the lithosphere). The initial configuration that Degen et al. (2025) use is of course not identical to that of ice & water. Yet, the features that occur are broadly similar.

The effect of mass redistribution and topography relaxation can be investigated using the setup 82 (line 83 of the parameter file) of figure 3, for both ASPECT and LaMEM simulations. Initially, the simulation is dominated by vertical motion during which the topography builds up (Figure 5). Very typical of this stage are very large vertical velocities which are >15 cm/yr in this case but can be up to meters/year in other initial configurations depending on the employed viscosity values. After a while, the velocity pattern is dominated by the geodynamic processes we are interested in. Only at this stage, results can be interpreted in a geodynamic manner. The ASPECT and LaMEM results are broadly consistent with each other (particularly given the much lower resolution employed in ASPECT).

Indeed, if we consider the vertical surface we see very large velocities during the initial stages, which become smaller at later model stages (Fig. 6), with somewhat similar patterns.

Most computational geodynamicists have observed this effect in simulations with a free surface. For example, in subduction zone simulations with a free surface, the typical subduction velocity patterns develop only after the topography of the continent has risen sufficiently. The isostatic adjustment stage is also discussed in a free surface benchmark study where all codes experience a quite similar trend with rapid initial uplift for case 2 (Crameri et al., 2012).

The duration of the isostatic adjustment stage is not the same in every situation. During a previous study together with T. Baumann, we performed probabilistic inversions for lithospheric rheological parameters to fit to the observed topography and surface velocity of the India-Asia collision zone (Baumann and Kaus, 2015). As in the current manuscript, different viscosity and density structures required a different waiting time until the isostatic balancing stage was over. At the time, we employed a heuristic criteria to distinguish the two stages (Baumann and Kaus, 2015); yet there is certainly room for additional (new) quantitative criteria for this. In absence of such criteria, one can either wait sufficiently long or manually check all simulations. The duration of the initial isostatic balancing stage largely depends on the employed viscosity and density structures, as well as on values employed in the sticky air approach. It is therefore not possible to indicate a generally valid waiting time; yet what helps is to start with an initially small timestep which is slowly ramped up until it reaches a maximum value. This is the recommended approach for LaMEM and used in the subduction examples for example, and is also suggested in the ASPECT documentation when they discuss free surface setups.

Some readers may wonder why we don't use models that start with an initial nonzero topography (say constructed following Airy isostasy). That is of course possible; yet a key assumption that underlies Airy isostasy is that lithospheric columns deform independently of each other – a situation that is only reached when the viscosity of the mantle and lithosphere is zero. The finite viscosity of the lithosphere on Earth results in a regional deformation if columns are moved upwards. In my experience, starting with an initial zero topography causes fewer small-scale velocity perturbations, but both situations can be performed.

[Figure]

**Figure 5.** Temporal evolution of a vertical velocity along a vertical cross-section through the middle of the model domain of model 82 (figure 3), with a corrected ASPECT setup that employs a free surface. Initially, the free surface (ASPECT) or internal free surface (LaMEM) rises, for the model to re-establish isostatic equilibrium. After some time, this transitions into a case where the surface topography is not dominated by isostatic equilibrium but by dynamic processes. Here this occurs somewhere between 100 and 250 kyrs. The transition between stage 1 (rebound) and stage 2 (dynamic processes) depends on the employed viscosity and density values and is not always clearly defined.

[Figure]

**Figure 6.** Temporal evolution of the surface velocity of the LaMEM and ASPECT models shown in Fig. 4. During the initial isostatic balancing stage, vertical velocities are very large which is an artefact of the model setup. Once the topography is roughly in isostatic balance, velocities are smaller. Note that surface velocities will only become zero once there is no dynamic motion more in the model (that is when all layers stratified with respect to their density and no far field stresses are applied – lithostatic limit).

Comparing our results as shown on Fig. 6 to those in the manuscript of Degen et al. (2025) (e.g. their Fig. 7, model 82) gives roughly consistent results and velocity patterns. The initial, non-balanced, stages are characterized by small topographies and large surface velocities. The "numerical instabilities" reported by Degen are therefore most likely not instabilities but the result of a non-balanced initial model setup along with terminating the simulation at a

very early stage. Running the model for a longer period of time (and with correct initial layer configuration) should resolve this.

True numerical instabilities can indeed occur in numerical modeling, as exemplified by the drunken sailor instabilities that may arise when explicit timestepping algorithms employ excessively large timesteps (Kaus et al., 2010). However, these instabilities are easily identifiable due to the abrupt sign reversal of the velocity field between consecutive timesteps. Fortunately, stabilization methods can enable the use of larger timesteps (Kaus et al., 2010; Rose et al., 2017), which is the default setting in LaMEM. The feature that Degen et al (2025) attribute to a "numerical instability" appears to be a density (or physical) instability instead, characterized by a relatively smooth evolution of the velocity with time which is correctly captured by both the (corrected) ASPECT and LaMEM simulations (as depicted in Fig. 6).

**Resolution tests**

The manuscript of Degen et al. (2025) lacks resolution tests. It's well-known that numerical models are approximations of the governing partial differential equations (Gerya, 2019). Therefore, it's crucial to ensure that the resolution is adequate for the processes being studied (Van Zelst et al., 2022). In the context of this paper, this involves verifying that the timestep is appropriate, that the grid cells are sufficiently small, and that the thickness and viscosity of the sticky air layer doesn't significantly impact the surface topography or velocity.

*Spatial resolution*

The LaMEM simulations used a fixed resolution of 96x96x128 staggered finite difference cells, which implies a vertical resolution of around 3 km in the crust (as no grid refinement was employed). Some of the lithospheric layers appear to be only 5-10 km in thickness, and depending on the setup, this may further thin with time (in some setups the mantle lithosphere has a lower density than the asthenosphere, resulting in thinning of crustal layers). As a rule of thumb, one needs around 5-10 gridpoints to resolve viscosity or density heterogeneities so the employed resolution is likely insufficient (Fig. 67 The ASPECT simulations use only 16 quadratic elements in the vertical direction with a regular grid spacing, which implies a 25 km resolution. Viscosities within a higher-order finite element must be smoothly varying to avoid numerical artefacts (Deubelbeiss and Kaus, 2008; Thieulot and Bangerth, 2025, 2022), which makes it certainly somewhat questionable whether the crustal layers in Degen et al. (2025) are appropriately resolved.

[Figure]

**Figure 7** Illustration of how the lower crustal layer is resolved in the LaMEM (left) and ASPECT (right) simulations for the models shown in Fig. 5 & 6. Note that the triangles are an artefact of the ParaView visualization as both codes use quadrilateral cells; the size of the elements in LaMEM is correctly represented, whereas the ASPECT output divides every quadratic element into 8 sub-elements for visualization purposes, so the actual finite element is twice coarser than what is shown here. LaMEM uses a marker and cell approach to track material parameters, whereas ASPECT employs a field-based advection approach in this setup.

Since Degen et al. (2025) only uploaded the marker files for LaMEM, but not the MATLAB or Julia scripts that generated them, I was not able to perform resolution tests with the data provided. For ASPECT cases this is in principle possible as ASPECT can use low resolution input grids as input for higher resolution simulations (e.g. using a piecewise constant interpolation method); yet, this would not appropriately resolve thinner crustal layers.

It should also be remarked that neither LaMEM nor ASPECT are ideal codes for the setup studied in this work, which involves sharp and sudden viscosity jumps and limited deformation. A higher numerical convergence rate is obtained with unstructured finite element codes that exactly mesh the viscosity jumps (Deubelbeiss and Kaus, 2008; Thieulot and Bangerth, 2025), which thus require fewer elements to resolve individual viscous layers.

*Temporal resolution*

The LaMEM setup uses a maximum timestep of 200 years, whereas the ASPECT free surface setup uses a timestep of 1000 years. No timestep convergence tests are shown in the manuscript of Degen et al. (2025). For this reason, I repeated the simulation using a setup in which the initial timestep of 100 years was allowed to ramp up until a maximum of 20kyrs which gave fairly similar results, suggesting that the authors could potentially speed up the simulations quite significantly (if this applies to other cases as well). Yet, it is a good idea to formally demonstrate this.

*Sticky air thickness*

The sticky air layer has been the focus of a benchmark study in which finite element and various codes with a sticky air layer were compared with each other (Crameri et al., 2012). The previous paper showed that the sticky air method gives reasonable results if it is sufficiently thick and has a sufficiently low viscosity. A non-dimensional parameter, $C_{isost}$, was suggested to be a useful criterion in this context:

$$C_{isost} = \frac{3}{16\pi^3}\left(\frac{L}{h_{st}}\right)^3 \frac{\eta_{st}}{\eta_l}$$

Here *L* is the width of the model (here the one of the uplifting lithosphere), $h_{st}$ the thickness of the sticky air layer, $\eta_{st}$ its viscosity, and $\eta_l$ the viscosity of the lithosphere. The analysis was done for the relaxation of a topographic perturbation and numerical simulations indicate a reasonably good agreement with numerical simulations when $C_{isost} < 1$. For the given model parameters indicated in this study (*L*~1200km, $h_{st}$=20km,   $\eta_{st} = 10^{18}$ Pas, $\eta_l$~$10^{20} - 10^{23}$Pas,   we obtain $C_{isost}$ =0.0131-13.1. In some cases, this is therefore likely fine, but in others perhaps not following this rule.

There are differences, however, between the LaMEM simulations and the earlier models used in the *Crameri et al. (2012)* study that employed free slip boundary conditions above the sticky air layer. LaMEM uses a stress-free upper boundary condition if the "open_top_boundary" option is activated (as is the case in the uploaded LaMEM input file). Such a stress-free condition allows for in- and outflow, which is less restraining than a layer with free slip top boundary (this effect is also clearly visible in Figure 4).  Accordingly, it is likely that satisfying results can be obtained for larger values of $C_{isost}$. A convergence test for the given setup using the extreme values of $\eta_l$ employed in this study will clarify this.

**Making data available & using open source software**

I commend the authors for uploading the input files of the forward models, which were crucial for reproducing the research results, even though the model description could have been clarified. This highlights the significance of making such data accessible in geodynamic studies, which will ultimately advance our community.

The reproduction of the results also highlights the fact that open source geodynamic codes cannot be used blindly but require verifications for the application at hand, along with some training in computational geodynamics.

ASPECT was developed from the onset to be an open-source community code and received funding for this purpose by the NSF through the Computational Infrastructure in

Geodynamics (CIG). LaMEM, on the other hand, was mostly developed at the University of Mainz for research applications and has not received dedicated funding to become a community code. Like ASPECT, many years of development have gone into it, and the code is available under an open source license (and certainly welcomes contributions or issues from users). Yet, given the difference in dedicated funding, the documentation of ASPECT is logically much more extensive than that of LaMEM.

As with any software, LaMEM may also have mistakes or bugs that haven't been identified yet. The normal way to deal with that in the open-source software community is to open an issue on the respective github/gitlab page to discuss it, such that other users of the software can also benefit from the discussion.

Lightly made and clearly unfounded claims about "numerical instabilities" of a particular software in a submitted manuscript (or even worse, a paper) are deeply disturbing, even if it is later claimed that the software may still be useful for the community (line 400 and onwards). If the authors of a manuscript still believe it is necessary to make such statements about software packages to which they did not contribute, the least they should do is make their utmost efforts to avoid mistakes on their end and double-check all results before submitting. Following the guidelines of good research practice of the German research foundation which appears to have funded this work (Deutsche Forschungsgemeinschaft, 2025), I would have expected *all four authors* to participate in verifying the numerical results. This would have made it highly likely that at least one of them had spotted that the ASPECT simulations were upside down.

**Summary**

Degen et al. (2025) presented new geodynamic model results that were used to train reduced-order models. I have shown that these results were incorrect due to inappropriate initial model configuration and in considering of isostatically non-balanced results. In particular:

a) The ASPECT simulations, which were used to demonstrate that LaMEM must have "numerical instabilities," use an initial model setup that is flipped upside down. If the initial geometry is corrected to match the LaMEM setup, the results of LaMEM and ASPECT are quite consistent, especially considering the much lower resolution used in the ASPECT simulation. This error in the model setup is easily detectable by simply viewing the *.vtu output files created by ASPECT.

b) Contrary to Degen et al. (2025), we don't observe numerical instabilities in the LaMEM forward simulations. Instead, the large vertical surface velocities observed during the early stages of the simulations are a natural and physical consequence of initiating

simulations with a flat initial topography. It takes time for such models to re-establish isostatic balance. Afterwards, the density differences between different lithospheric layers and the asthenosphere induce further surface deformation, leading to dynamic topography changes. Consequently, the initial, rapid vertical uplift stage should not be interpreted geodynamically because it results from a highly artificial initial configuration. The duration of this initial isostatic balancing stage varies among simulations and is particularly sensitive to the viscosity of the setup. Therefore, verification of the model output is necessary to ensure that one is beyond the initial isostatic adjustment phase.

c) Degen et al. (2025) do not specify the time point at which they extract the surface velocities from their forward LaMEM simulations. Examining their figures strongly suggests that in certain instances, they rely on results from cases that are not yet in isostatic balance (refer to their figure 6, for instance).

d) No resolution tests are presented. Since some of the crustal layers appear to be relatively thin compared to the vertical resolution of the LaMEM/ASPECT simulations, they may be unresolved. Additionally, resolution tests for the sticky air and the timestep are lacking.

e) While trying to reproduce their models, I noticed that there is significant confusion about the parameters used in the simulations as shown in the figures and the uploaded data files, because the employed model numbers are not listed in the parameter tables.

f) ASPECT input files as provided do not work with version 2.5.0 of ASPECT.

The idea of utilizing the output of geodynamic simulations to train reduced-order models, which is the focus of this manuscript, remains valid. However, it is crucial to use scientifically meaningful forward modeling results as input for this workflow. The forward models employed in the current version of the manuscript thus require either redoing or a careful re-analysis. In its current state, the paper cannot be accepted. However, with significant additional work and rigorous resolution tests, it could become a valuable scientific contribution.

**Open data**
The input scripts for the LaMEM and ASPECT simulations (corrected such that they work with ASPECT v2.5.0) are made available in a permanent Zenodo repository (Kaus, 2025).

**References**

Baumann, T.S., Kaus, B.J.P., 2015. Geodynamic inversion to constrain the non-linear rheology of the lithosphere. Geophys. J. Int. 202, 1289–1316. https://doi.org/10.1093/gji/ggv201

Crameri, F., Schmeling, H., Golabek, G.J., Duretz, T., Orendt, R., Buiter, S.J.H., May, D.A., Kaus, B.J.P., Gerya, T.V., Tackley, P.J., 2012. A comparison of numerical surface topography calculations in geodynamic modelling: an evaluation of the 'sticky air' method: Modelling topography in geodynamics. Geophysical Journal International 189, 38–54. https://doi.org/10.1111/j.1365-246X.2012.05388.x

Degen, D., Kumar, A., Scheck-Wenderoth, M., Cacace, M., 2025. Exploiting Physics-Based Machine Learning to Quantify Geodynamic Effects – Insights from the Alpine Region. EGUsphere 1–24. https://doi.org/10.5194/egusphere-2025-1925

Deubelbeiss, Y., Kaus, B.J.P., 2008. Comparison of Eulerian and Lagrangian numerical techniques for the Stokes equations in the presence of strongly varying viscosity. Physics of the Earth and Planetary Interiors 171, 92–111. https://doi.org/10.1016/j.pepi.2008.06.023

Deutsche Forschungsgemeinschaft, 2025. Guidelines for Safeguarding Good Research Practice. Code of Conduct. https://doi.org/10.5281/zenodo.14281892

Heister, T., Dannberg, J., Gassmöller, R., Bangerth, W., 2017. High accuracy mantle convection simulation through modern numerical methods – II: realistic models and problems. Geophysical Journal International 210, 833–851. https://doi.org/10.1093/gji/ggx195

Kaus, B., 2025. Corrected LaMEM and ASPECT input files for Degen et al. (2025), EGUsphere 1–24. https://doi.org/10.5281/zenodo.15828911

Kaus, B.J., Popov, A.A., Baumann, T., Pusok, A., Bauville, A., Fernandez, N., Collignon, M., 2016. Forward and inverse modelling of lithospheric deformation on geological timescales, in: NIC Series. Presented at the NIC Symposium 2016, pp. 299–306. https://doi.org/2128/9842

Kaus, B.J.P., Mühlhaus, H., May, D.A., 2010. A stabilization algorithm for geodynamic numerical simulations with a free surface. Physics of the Earth and Planetary Interiors 181, 12–20. https://doi.org/10.1016/j.pepi.2010.04.007

Kumar, A., 2025. LaMEM and ASPECT input and data files corresponding to Exploiting Physics-Based Machine Learning to Quantify Geodynamic Effects – Insights from the Alpine Region. https://doi.org/10.5281/zenodo.15478977

Rose, I., Buffett, B., Heister, T., 2017. Stability and accuracy of free surface time integration in viscous flows. Physics of the Earth and Planetary Interiors 262, 90–100. https://doi.org/10.1016/j.pepi.2016.11.007

Thieulot, C., Bangerth, W., 2025. On the choice of finite element for applications in geodynamics – Part 2: A comparison of simplex and hypercube elements. Solid Earth 16, 457–476. https://doi.org/10.5194/se-16-457-2025

Thieulot, C., Bangerth, W., 2022. On the choice of finite element for applications in geodynamics. Solid Earth 13, 229–249. https://doi.org/10.5194/se-13-229-2022

---

## Author Response (AR1)

**Answer to Reviewer 1 (Sergio Zlotnik)**

Thank you very much for taking the time to review our paper and helping us to improve its current state. Below, we outline how we will address the comments in the manuscript itself.

- Section 2.1: is the forward model linear? or the viscosity \eta is a function of velocities \bf{u}? This is very relevant to the discussion of section 4.1.1 (errors for eta are smaller that for rho)
    - We solve for a linear forward Stokes Flow problem, that is, we do not consider any functional behavior of the effective viscosity parameter either in terms of solid velocities nor strain rates. This information has been added to section 2.1 in line 67 and lines 80-81.
- Section 2.2: Are the snapshots centered before the svd? Number of snapshots? Size of the test set? This is explained later, but maybe it can be repeated in 2.2.
    - Reviewer 1 is right: the snapshots are centered before executing the svd. We added all required information to section 2.2 in lines 104-107 and line 110.
- Section 4.1.1: How local and global errors are measured? It is stated that errors are of order of 1m (tol 1E-4) although in Fig 2 errors are 1 to 5%. How should I interpret these results?
    - Global errors are calculated as the MSE of the back-projected solution. We added the exact formula in lines 112-114. The problem with global errors is that, locally, the error can exceed these values. Therefore, we additionally investigated the local error behavior (by taking the difference between the full- and reduced order solution) for designated solutions. This we discussed in Fig. 2, where we demonstrate how the maximum local difference is an order of magnitude higher than the global error. We added a more detailed description in lines 205-210 to better highlight the differences between these two error metrics and their importance for the evaluation of the approximation quality.
- I could not follow the discussion on the viscosity parameterization. Are there 6 or 2 viscosity parameters?
    - The forward model has six viscosity parameters. However, during the construction of the surrogate model, we decided to combine some layers in terms of their viscosity parameterization, and consider only the viscosity of the lithosphere (including slabs) and asthenosphere as free parameters. We add some clarifying text in lines 219-222.
- "However, as we will detail in subsection 4.3, this surrogate model is currently not constructible because of challenges related to the data set." why?
    - The reason behind our statement is that, during the construction of the surrogate model for the case where we combined both density and viscosity, we did find that some of the full order forward solutions showcased velocities and topographies that vary by several orders of magnitude with respect to the other family of solutions. Worth to mention is that these were solutions to fully-solved forward simulations (that is we did encounter any crash or premature ending of the forward model) and that they appear to follow a random pattern within the parameter space. The presence of random variations, that is, solutions that do not follow the expected physical behavior, hindered to construct a surrogate model. While addressing the comments received during the first round of review, community post and reviewer

2, we did carry out an in-depth revision of the workflow to extract the results from the full order solutions to construct the surrogate model and we realized that the appearance of these random „unexpected" solutions was due to an error in the shell script we use to identify the timestep from where to extract the full order solutions (for those cases it was picking an earlier timestep before the system was fully isostatically balanced). We would like to stress here that we double check all three scenarios discussed in the manuscript and we found that this error only occurred for scenario 3, and only for few realizations within this scenario, being the reason, we did not spot the problem before. Given that we identified the source of the problem, we carried out the construction of the surrogate model for this last scenario as well, which is presented in section 4.3 and in Figure 2.

- The use of POD is fairly clear. But for reproducibility it would be necessary that the configuration of the NN is included in the manuscript or in the public data.
    - The main hyperparameter of the NN are listed in the tables of Figure 2, additionally we provided a data folder containing the scripts for the surrogate model construction of all surrogate models used in the paper. The "SurrogateModelScripts" folder has always a file which contains the configuration of the NN. To better highlight this, we added a description in lines 178-180.

**Answer to Reviewer 2 – Frist Round**

We would like to thank reviewer 2 for his/her comments.

We would like to emphasize, that the main objective of our study is rather on describing a novel methodology to construct surrogate models for geodynamic simulations and to allow for probabilistic inverse studies and not on the respective geodynamic simulation methods that the surrogate model will be applied to. However, the comments from reviewer 2 are about aspects related to some of the forward simulations performed with the open-source software LaMEM and only applied to the last scenario presented and discussed, scenario 3. Thus, the comments of reviewer 2 exclusively focus on aspects that make up only a small portion of the entire work. We thank reviewer 2 for these comments that helped together with the community post to identify the issues with the last scenario and want to elaborate with some detail to those: As a matter of fact, the methodology presented in the manuscript is not dependent on the forward dynamic solver adopted (as illustrated in section 5). The problems encountered while constructing the surrogate for this last scenario do not impact the scientific merit of the results in principle and we added the surrogate models for the combined scenario. As detailed in our answer, we confirm that there was an error in the extraction of the timesteps and we acknowledge all comments raised from reviewer 2, whose expertise with the specific software has helped improving parts of the applications presented. Having identified this error in the script we used to extract the corrected timesteps, and having realized that the error only affected a few simulations for the last scenario, we have now constructed a surrogate model for this last case in the revised manuscript. Let us disclaim here that the problems encountered, wrong timestep extraction, are solely related to the combined scenario.

Please find below a description of how we addressed the comments in the paper.

- The development of surrogate models represents a noteworthy and valuable contribution. However, it is important to express concerns regarding the reliability of the underlying forward models. Several simulations terminated prematurely due to a crash in a third-party library

(details are provided in Appendix A). These failed simulations were not correctly identified and were instead interpreted as successful, albeit exhibiting "instabilities" such as low topography and high vertical surface velocities. In reality, these anomalies are attributable to the premature termination of the simulation, which prevented the system from reaching isostatic equilibrium. Substituting the crashing solver component with an alternative resolves the issue entirely.

- First of all, we would like to clarify that all our simulations did not „terminate prematurely" and we did not face any problem with any specific „third-party library". The source of the problem stems from an error in the automatic extraction of the results through a shell script. All results were supposed to be extracted at ~0.27 Myr (a clarification has been added in lines 148-151), a time when the system is in isostatic balance. However, the error in the shell script leads to an occasional extraction after 0.027 and 0.0027 Myr. Because of the random nature of this occurrence, we did not spot the problem before. Therefore, we cannot agree with the conclusive remark from reviewer 2 stating that the source of errors stems from solvers failure. As already stated in our general comment above, all results from the forward simulations have been already stored, so there is no need to carry out additional „full-order simulations".

- A general note of caution here: the problem rarely lies on the library (MUMPS), rather on the way it is used. MUMPS is a well-known and widely used distributed-memory parallel solver (the same apply to superLU_DIST). The reason for us to prefer to rely on MUMPS instead of a more frontal solver as superLU_DIST stems mainly from the former being designed for a relatively broader range of matrices and relying on a dynamic pivoting over hybrid parallelization. This said, it is relatively hard for us to pin-point a likely potential reason for the crashing simulations experienced by reviewer 2: we had too few details on the configuration, architecture, library versions and similar aspect on his/her local installation (a configuration log would help us in this regard) to be able to reproduce the crashing simulation.

- Although the affected simulations are explicitly reported only for the combined training set, it cannot be definitively ruled out that similar problems may exist in the other sets. Therefore, it is strongly recommended that all forward simulations be recomputed to ensure consistency and reliability. As this task exceeds the scope of a standard major revision, the manuscript should be reconsidered for resubmission after the complete regeneration of the training datasets.

  - We double checked the extraction regarding all data sets and can confirm that the issue is solely related to the combined scenario 3, where we changed both the density and viscosity. Given that the error is not related to the forward solution, but rather to a wrong extraction of the results, no further forward simulations are required to generate the training datasets for the last scenario. The training data sets for scenario 1 and 2 do not require any modification either.

  - The newly generated surrogate models are presented in section 4.3 and Figure 2 has been updated to account for the additional surrogate models. We also updated Figure 7 correcting for the previous wrongly extracted forward solves. In light of these changes also the Discussion has been updated, especially in lines 364-368 and 424-444 and the conclusion was adapted in lines 464-472.

- To reduce the computational cost associated with regenerating these datasets, an optimized set of input parameters and solver configurations may be employed. Appendix B summarizes a series of such optimizations, which together yield a reduction in run time by more than two orders of magnitude.

  - Thank you for the suggestions. However, as stated above, all simulation results for all scenarios, including scenario 3, have been previously stored. Therefore, there is no need

to rerun any simulations, but rather to redo the extraction step. A reference to the revised version of the data set is provided in the revised manuscript.

- Rather than including the datasets themselves, the supplementary material should provide scripts used to extract relevant information from the simulation outputs. It is also recommended to include automation scripts for input file generation and forward model execution. Appendix C presents an example Python implementation of this functionality.
  - As pointed out in the general comment, the main objective of the paper is to leverage surrogate model construction for geodynamical simulations to enable probabilistic inverse methods in this field. Therefore, the data folder concerning the surrogate models was constructed with this main aim in mind. To ease the surrogate model construction, the simulations results are provided in form of data sets. Our decision was motivated mainly by the wish to enhance ease, from the user side, to reproduce the surrogate model construction and its applications (the main subject of the manuscript) without involving any installation of the forward solver used for generating the training set. Nevertheless, to ease the reproducibility of the forward model, we included the mentioned shell script in the data folder containing the information about the forward model (see second round of comments).
- Detailed comments
  - These detailed comments relate to specific aspects regarding the combined scenario. Because of the above detailed reasons, we completely reworked section 4.3 and parts of the discussion in the revised manuscript. Hence, the detailed comments refer to text parts that are no longer existing in the document.

**Answer to Reviewer 2 – Second Round**

We would like to thank reviewer 2 for his/her comments.

On a general note, we would like to stress again the focus of the study, that is, to showcase a workflow to construct surrogate models that conform to the actual physics at play in order to enable setting up and solving inverse problems as applied to typical geodynamic simulations.

This said, we notice that, unfortunately, the comments raised by reviewer 2 again do focus on a particular aspect related to a subset of forward full-order geodynamic simulations, but do not address the other scientific aspects of the study. While we already stated that, thanks to the community post as well as the first round of comments raised by reviewer 2, we have been able to identify the source of errors for some forward simulations (due to a wrong timestep selection during the postprocessing stage), we would like to clarify that the error made relates to a minor part of the study (a single sub-set of the simulations) and should not be considered as downgrading the scientific merit of the study, which stems from the surrogate model construction and its usability for geodynamic applications. As stated in our first round of replies, we provide a revised manuscript where the surrogate model is constructed also for the final subset of simulations, and we hope that with the new addition, the message and merit of the study is clarified.

In what follows, we provide a point-by-point answer to all comments raised during the second round of posting.

- "First of all, we would like to clarify that all our simulations did not „terminate prematurely" and we did not face any problem with any specific „third-party library."

To allow an independent assessment of the correctness of this statement, please upload the log files of all forward simulations.

In addition to the log files, please provide the complete software infrastructure used to accomplish the following tasks:

A) Preparation of the input files for each realization

B) Submission of each realization using a workload manager (e.g., slurm)

C) Error checking for forward model runs (this part is absolutely critical).

D) Extraction of input data for surrogate models from forward simulation outputs

According to GMD policy, all of these critical software components must be made publicly available.

- o We can agree only partly with the point raised by reviewer 2. We entirely agree with the „FAIR" principles of code and data as per the journal policy, but found some of the requests from reviewer 2 not consistent with the GMD data policy. To be more specific, we agree with the majority of requests, but find point (B) puzzling, which relates to bash scripting for the slurm scheduler (or any other scheduler) used. We personally do not see how such additional information would adhere to the „FAIR" principles, since job submission depends on the particular architecture of the cluster used (and this also entails available libraries and modules) as well as the running scheduler. How would it help to judge the scientific merit of the manuscript?

- o Concerning all other points, we uploaded all input files used to run the simulations (point A), which are also informative of the error criteria adopted in each run (point C). In addition, the input data used to construct the surrogate models (currently present in the paper) has already been stored in a dedicated repository. The surrogate models that are provided in addition along with their data sets have been added to the data repository.

- o To clarify the working procedure, we rely on bash shell script "make_models.sh" to prepare the forward full order model [**https://doi.org/10.5281/zenodo.16640814**], with a combination of "awk" to read the input parameters from the comma delimited files, and a stream editor "sed", to edit the material input properties in the LaMEM input file. Both of these, awk and sed, are part of Unix/Linux distributions. The shell script produces directories for each model containing a LaMEM input file and a slurm script to submit the job, which are provided here [**https://doi.org/10.5281/zenodo.16640814**].

- o We ran the simulations until a minimum of 0.3 Myr since we fixed the time step at which data is extracted for the surrogate model (see below). For this, we relied on another shell script, "time_step_use_find.sh, which lists the time step within a given range of interest (0.2-0.3 Myr). This script produces an output text file with the model name (first column) and the associated time-step directory names (following columns). We then used this output file to read the simulation results in Paraview and write them out as .csv files using a customised Python script ("save_time_step.py" [**https://doi.org/10.5281/zenodo.16640814**]). For the models for which those timesteps were missing, we checked the simulation status in the slurm log file, and, in case of premature failure, non-convergence of the solvers, we adjusted the solver settings related to the number of iterations and the multigrid solver options [solver_type_2.dat,**https://doi.org/10.5281/zenodo.16640814**]. All this information is provided within the respective input files [**https://doi.org/10.5281/zenodo.16640814**]. To avoid any confusion, we have now

        added all input files to the data repository instead of providing only one exemplary input file (as done during the initial submission).

- *"The source of the problem stems from an error in the automatic extraction of the results through a shell script. All results were supposed to be extracted at 0.27 Myr, a time when the system is in isostatic balance. However, the error in the shell script leads to an occasional extraction after 0.027 and 0.0027 Myr."*

  This error appears rather puzzling, especially considering that your simulations do not output data at any of the mentioned time steps (0.27, 0.027, or 0.0027 Myr). The closest timestamps I found in the output logs are: 2.796436e-01, 2.964359e-02, and 2.843589e-03. Furthermore, there is no apparent pattern in the problematic realization numbers (82, 89, and 95), which suggests the issue occurs randomly. Please provide a detailed explanation of this bug, along with the actual extraction script.

  - We realised that this point requires further clarification. The pattern in the time step we do reference (0.27-0.0027 Myr) is to illustrate the issue. Indeed, the actual timesteps from the output list might follow a similar order, as for example extracted from a random solution:
    Timestep_00000100_2.28830780e-02
    Timestep_00000620_2.82883078e-01

  - As shown in the example, the naming convention lists the time step number as the first number in the file name. Afterwards, the value of the timestep is printed, followed by the exponent power as the last number. Hence, this exponent power dictates the extraction, whether we are at 0.28 or 0.028 Myr.

  - To illustrate the issue we encountered, we show below the previously used script for the extraction (in bold the relevant lines):
    *#!/bin/bash*
    *rm -f use_time_Step # file to store model name and time step directory; remove if it #exists*
    *touch use_time_Step # create the file*
    *for dir in Train_* Validate_* # iterate over all the models*
    *do*
    *cd ./$dir # go inside the directory and run following*
    **echo $dir `ls -d Timestep_0000*_2.[6-9]*`>>./../use_time_Step # print model name #and listed directory with the prescribed pattern**
    *cd ./../*
    *done*
    This script listed all the timesteps with the above pattern, from the lowest exponent (e-03) to the highest in ascending order. The issue arises from the circumstance that the first entry in the time-step column is read by the Python script in Paraview. Meaning that for some realisations, the timestep with the wrong exponent has been considered. As stated, we noticed an error in the script, which we have now corrected to:
    echo $dir `ls -d Timestep_0000*_2.[6-9]*e-01`>>./../use_time_Step
    This ensures that the data is picked at the desired time step.

- Why was 0.27 Myr chosen? This duration is not sufficient to reach isostatic balance. Based on your typical model results, I estimate that at least 0.5 Myr is required. To remain on the safe side, 1 Myr should be used. Alternatively, I suggest implementing a custom termination criterion in LaMEM based on a surface velocity threshold. Therefore, I strongly recommend

that all forward models be recalculated for a longer simulation period (at least 0.5 Myr, preferably 1 Myr).

- o The aim of the paper is the construction of surrogate models to enable a better constraint of the present-day dynamic state of the lithosphere and mantle in a manner which is consistent with available observations (e.g, Topography and GNSS velocities), and to investigate in future studies the sensitivity of the results, via global sensitivity analysis, to parameters variations due to inherent uncertainties. Hence, we are interested in the quasi-instantaneous dynamic response of the system to the internal deep mass distribution, mantle and slabs.
- o Our choice of considering an elapsed time of 0.27 Myr and not 1 Myr is based on the following criteria:
  - ▪ 1) The Stoke flow induced by the considered slabs is established in the model and does not deform significantly (Fig. 1);
  - ▪ 2) Changes in the topography and surface velocities do not vary significantly. This threshold time is decided by observing the change in variation of topography and velocities for the reference model (Fig. 1).
- o After ~0.2 Myr change in mean topography is in the order of ~10 m and < 1mm/yr, respectively. These changes are well within the resolution of digital elevation models and GNSS-derived velocities. Hence, we chose a threshold time of ~0.27 Myr, to also account for variations in physical properties to be on the safe side. An explanation has been added in lines 148-151.

[Figure]

*Figure 1: Depth slice of the viscosity at 220 km showing the geometry of the slab for the reference model (higher viscosity:red color) with time (a-d). Flow is shown by arrows scaled by magnitude. Note the decreasing size of the slab region with time.*

- o Worth to note is that considering a longer time-period, would have induced active deformation within the slabs, leading to a system geometry (and mass distribution) no longer representative of the present-day architecture of the mantle (the one we

imposed at the beginning of the model), which would enter systematic (epistemic) bias in the later sensitivity analysis where the actual configuration would have been dependent on the applied material properties, thus hindering an objective comparison of the different response only in terms of the varying properties.

[Figure]

*Figure 2: Time evolution of the mean absolute topography (left) and mean absolute vertical velocity (right) change for the reference model. Note that for the velocity panel y-axis is clipped for the initial time to visualise the later variation.*

- Additionally, I recommend increasing the spatial resolution, as some geological units appear to be underrepresented. Ideally, a resolution sensitivity test should be included.
    - Regarding the resolution tests and increasing the resolution of the models, we do not agree with the comments raised with respect to the specific objective of the study. In our study, we target the deeper mantle configuration only, and therefore, we consider that the current resolution (~13 km in E-W, ~17 km in N-S, and ~3 km along depth) is enough to reflect the typical wavelength of the expected response within constraints from geophysical/geological data. For example, at this resolution, the influence of the lithospheric thickness, upper-mantle architecture derived from the different tomography models (i.e., effects of slab), and the local sensitivity analysis of their physical properties (density and viscosity) are well captured as discussed in Kumar et al 2022 (https://doi. org/10.1029/2022GL099476). We would agree with the comment raised if our study had attempted to further investigate the source of the smaller wavelength response, as due to internal variations in the crustal configuration, which is fixed. Furthermore, the main focus of the paper is the construction of the surrogate models, and the structure of the model is not changed; rather, the physical properties are varied.

- There is no need to upload gigabytes of extracted data, as this can easily be regenerated. Of course, uploading it remains optional.
    - That seems to be a misunderstanding of our previous answer. We wanted to confirm that the data sets are still on our internal servers and thus allow us to redo the extraction step. However, we see no added value in uploading all output files, since, as pointed out by the reviewer, they can be easily regenerated.

- In any case, the complete software infrastructure related to the forward modeling process must be shared in accordance with GMD policy.

- We have shared the scripts used to generate the input files and the data extraction. In addition, we also share the input files for all the models such that they can be reproduced. Please also refer to our previous answer to Comment 1 by the same reviewer.